nanotechnology

ferroelectric, superlattice, perovskite

**Author for correspondence:**
Daniel Bennett
e-mail: db729@cam.ac.uk

# Electrostatics and domains in ferroelectric superlattices

Daniel Bennett[1], Maitane Muñoz Basagoiti[2,3,4]
and Emilio Artacho[1,4,5]

[1]Theory of Condensed Matter, Cavendish Laboratory, Department of Physics,
J J Thomson Avenue, Cambridge CB3 0HE, UK
[2]Faculty of Science and Technology, University of the Basque Country, Barrio Sarriena 48940 Leioa, Spain
[3]Gulliver Lab UMR 7083, ESPCI PSL Research University, 75005 Paris, France
[4]CIC Nanogune and DIPC, Tolosa Hiribidea 76, 20018 San Sebastian, Spain
[5]Ikerbasque, Basque Foundation for Science, 48011 Bilbao, Spain

DB, 0000-0003-0892-2125; EA, 0000-0001-9357-1547

The electrostatics arising in ferroelectric/dielectric two-dimensional heterostructures and superlattices is revisited within a Kittel model in order to define and complete a clear paradigmatic reference for domain formation. The screening of the depolarizing field in isolated ferroelectric or polar thin films via the formation of 180° domains is well understood, where the width of the domains $w$ grows as the square-root of the film thickness $d$, following Kittel's Law for thick enough films ($w \ll d$). For thinner films, a minimum is reached for $w$ before diverging to a monodomain. Although this behaviour is known to be qualitatively unaltered when the dielectric environment of the film is modified, we consider the quantitative changes in that behaviour induced on the ferroelectric film by different dielectric settings: as deposited on a dielectric substrate, sandwiched between dielectrics, and in a superlattice of alternating ferroelectric/dielectric films. The model assumes infinitely thin domain walls, and therefore is not expected to be reliable for film thickness in the nanometre scale. The polarization field $\mathbf{P}(\mathbf{r})$ does vary in space, deviating from $\pm P_S$, following the depolarizing field in linear response, but the model does not include a polarization-gradient term as would appear in a Ginzburg–Landau free energy. The model is, however, worth characterizing, both as paradigmatic reference, and as applicable to not-so-thin films. The correct renormalization of parameters is obtained for the thick-film square-root behaviour in the mentioned settings, and the sub-Kittel regime is fully characterized. New results are presented alongside well-known ones for a comprehensive description. Among the former, a natural separation between strong and weak ferroelectric coupling in superlattices is found, which depends exclusively on the dielectric anisotropy of the ferroelectric layer.

# 1. Introduction

The formation of ferromagnetic [1–4] and ferroelectric [5,6] domain structures in thin films is a well-known phenomenon. Polydomain structures appear in ferroelectric thin films in order to screen the electric depolarizing field arising at the interfaces between the surfaces of the thin film and its environment, such as vacuum or a non-metallic substrate. The electrostatic description of a ferroelectric thin film in an infinite vacuum has been studied in detail [6,7]. The equilibrium domain width $w$ follows Kittel's Law versus film thickness $d$, $w \propto \sqrt{d}$, when $w \ll d$. Within the same model but making no approximations on the electrostatics arising from an ideal, regular polydomain structure, for $w \gtrsim d$, $w$ reaches a minimum and grows again when decreasing $d$, until the monodomain is reached [6,7]. A similar effect was first predicted and observed in ferromagnetic thin films [8–11]. This description of an isolated thin film does not describe the effect that the surrounding environment has on the electrostatics of the thin film and hence the domain structure, however.

It is now possible to fabricate ferromagnetic and ferroelectric samples by growing alternating layers of different thin films, just a few unit cells in thickness, in a periodic array (superlattice) [12–14]. Alternating between ferroelectric and paraelectric layers (FE/PE superlattice, see figure 1), a great deal of control over the superlattice's properties can be achieved by changing the relative thicknesses of the layers [15–18]. This has generated interest in the study of FE/PE superlattices from the theoretical [19,20] and computational [21] perspectives.

The dependence of the domain structure on superlattice geometry cannot be described using the theory of a thin film in an infinite vacuum, however. Some generalizations have appeared in the literature which include the effects of surrounding materials [19,22–28]. For a free-standing thin film on a substrate, it was claimed that the electrostatic description is the same as for a thin film of half the thickness sandwiched between two paraelectric media [22]. This has been used to fit measurements of ferroelectric domains [29,30], but a free-standing film on a substrate was never studied explicitly.

By placing a ferroelectric thin film together with a paraelectric layer between two short-circuited capacitor plates, it was found that the domain structure could be controlled by tuning the properties of the paraelectric layer, and the stability of the ferroelectric film could be improved [23–28]. This system is to some extent equivalent to a FE/PE superlattice since the capacitor plates impose periodic electrostatic boundary conditions.

A number of experimental and computational advances have revived interest in this problem. Interesting effects can occur at interfaces between different materials such as the formation of a two-dimensional electron gas (2DEG) at polar/non-polar interfaces like $LaAlO_3$/$SrTiO_3$ (LAO/STO) [31,32]. It is thought that the 2DEG appears to screen the polar discontinuity at the LAO/STO interface [33], and similarly, it has recently been proposed as a mechanism to screen the depolarizing field at ferroelectric/paraelectric interfaces [34,35]. This is difficult to directly observe experimentally, and evidence for 2DEG formation at FE/PE interfaces has only very recently been found [36–38]. This is because there is competition with domain formation for the screening of the depolarizing field. Since these phenomena are of an electrostatic origin, a clear picture of the electrostatics of ferroelectrics is essential in order to understand them.

Although ferroelectric thin films have been frequently simulated from first principles in different settings and environments [35,39–43], ferroelectric domains are quite demanding to simulate from first principles, as they require much larger supercells. Recent developments in effective model building from first-principles calculations (second-principles methods) make it possible to study very large systems, including large domain structures in ferroelectric materials [44–52] and observe interesting related effects such as negative capacitance [53] and polar skyrmions [54]. These scientific advances, both experimental and computational, have motivated us to revisit the electrostatic description of ferroelectric domains.

The continuum electrostatic description of a monodomain ferroelectric thin film is essentially unaffected by a dielectric environment of the film. This is because there is zero field outside the thin film and hence these regions make no contributions to the electrostatic energy. For a polydomain ferroelectric thin film, the domain structure introduces stray electric fields into the regions outside the film (figure 2). We expect different behaviour if we replaced the vacuum regions with a dielectric medium. Understanding the effect of more general geometries on the electrostatic description of ferroelectric thin films not only gives an insight into how the surrounding dielectric media contribute to the screening of the depolarizing field, but also allows us to understand the behaviour of the domain structure of the film in different environments, bringing us closer to a realistic description of a thin film. Here, we present the results for a Kittel model for domains in ferroelectric films and superlattices in different electrostatic settings. We present known results together with new ones for a comprehensive, comparative description of the following situations:

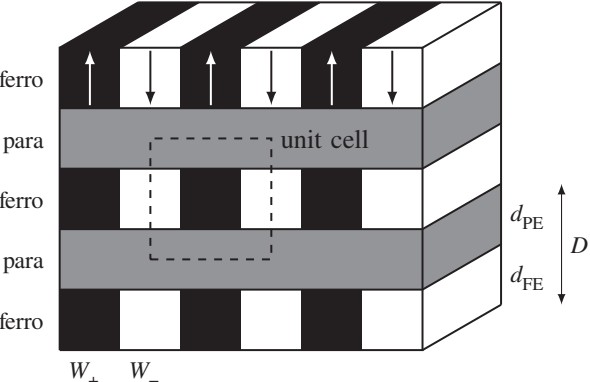

**Figure 1.** Geometry of a FE/PE periodic superlattice. The unit cell is indicated by the dashed square. The thicknesses of the layers are indicated on the right and $W_+$ and $W_-$ are the widths of the positive and negative polarization domains. In polydomain limit, these widths are equal: $W_+ = W_- = w$.

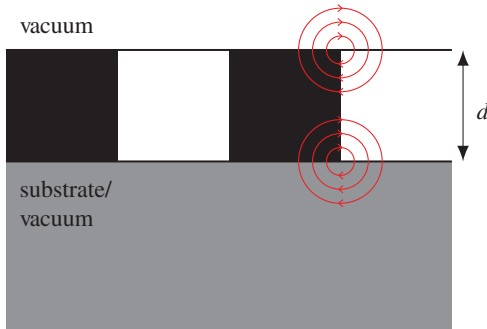

**Figure 2.** Geometry of a ferroelectric thin film of thickness $d$ with a 180° polydomain structure. The red lines represent the electrostatic depolarizing field, which bend around the interfaces and domain walls.

First, we review the continuum model of an isolated film (IF) with the full treatment of the electrostatics and a domain wall term. We then generalize the theory for three different systems: a thin film on an infinite substrate (overlayer, OL), a thin film sandwiched between two infinite dielectric media (sandwich, SW), and a FE/PE superlattice (SL). We keep the prevalent nomenclature in the literature of referring to a spacer material such as STO as paraelectric, but the description will be exclusively that of a dielectric material with a given isotropic dielectric permittivity.

All of these systems except the OL have appeared in the literature in various contexts and with different levels of detail. We compare the different cases, first in the Kittel limit ($w \ll d$), for which analytic expressions are obtained for $w(d)$, and also in the general situation. Previous studies of periodic superlattices have assumed ferroelectric and paraelectric layers of equal width. Here, we provide a more general study of domain structures as a function of superlattice geometry. We also present a detailed derivation of the electrostatic energies in appendix A.

## 2. Review of model for a film in vacuum

The fundamental model used in this work is based on the following free energy *per unit volume* of a ferroelectric thin film in a vacuum with a 180° stripe domain structure [1,5]

$$\mathcal{F} = \mathcal{F}_0(P) + \frac{\Sigma}{w} + \mathcal{F}_{\text{elec}}(w, d), \tag{2.1}$$

where $\mathcal{F}_0(P)$, defined as

$$\mathcal{F}_0(P) = \frac{1}{2\varepsilon_0 \kappa_c} \left( \frac{1}{4} \frac{P^4}{P_S^2} - \frac{1}{2} P^2 \right) \tag{2.2}$$

is the bulk ferroelectric energy with spontaneous polarization $P_S$ and dielectric permittivity $\kappa_c$, which describes the curvature about $P = P_S$. $\Sigma$ is the energy cost per unit area of creating a domain wall,

$\mathcal{F}_{\text{elec}}$ is the electrostatic energy associated with the depolarizing field, and $w$ and $d$ are the width of one domain and thickness of the film, respectively.

In the Kittel model, instead of solving for $P$ in equation (2.2), the total polarization field $\mathbf{P(r)}$ is taken to deviate from the spontaneous polarization $\pm P_S$ in linear response to the electric depolarizing field, according to the dielectric susceptibilities normal and parallel to the film, $\kappa_c$ and $\kappa_a$, respectively. This model makes significant approximations about the form of $\mathbf{P(r)}$, such as neglecting domain-wall width and surface/interface effects. Ferroelectric domain walls tend to be much thinner than ferromagnetic domain walls, typically of order 1 nm. Realistic descriptions of nanometric films should rather resort to theories with proper consideration of those effects, such as explicit first-principles calculations or Ginzburg–Landau (e.g. [19,20,55]). There are situations, however, for which this model is relevant (in our case, this work was prompted by situations as described in [34,56]), and, more generally, a clear account for the behaviour of this simple model in the electrostatic settings considered represents a valuable paradigmatic reference.

Since we will be interested in the electrostatic effects due to a finite polarization, we will consider the polarization to be $P_S$, except for its modification in linear response to the depolarizing field implicit when using a dielectric permittivity for the material normal to the field, $\kappa_c$. This assumption is equivalent to replacing the form of $\mathcal{F}_0(P)$ in equation (2.2) by its harmonic expansion about one of the minima

$$\mathcal{F}_0(P) = \frac{1}{\varepsilon_0 \kappa_c}(P - P_S)^2. \tag{2.3}$$

The equilibrium domain structure for this system for a given thickness is obtained by minimizing the energy: $\partial_w \mathcal{F} = 0$.

As mentioned above, we consider an ideal domain structure made by regular straight stripes, all of them of the same width $w$ (in appendix A different widths are considered). For an IF, the electrostatic energy for that structure is given by [6]

$$\mathcal{F}_{\text{elec}} = \frac{8 P_S^2}{\varepsilon_0 \pi^3} \frac{w}{d} \sum_{n \text{ odd}} \frac{1}{n^3} \frac{1}{1 + \chi \kappa_c \coth((n\pi/2)\chi(d/w))}, \tag{2.4}$$

where $\kappa_a$, $\kappa_c$ are the dielectric permittivities in the directions parallel and normal to the film and $\chi = \sqrt{\kappa_a/\kappa_c}$ is the dielectric anisotropy of the film. In the Kittel limit [1,5], $w/d \ll 1$, equation (2.4) reduces to

$$\mathcal{F}_{\text{elec}}^{\text{Kittel}} = \frac{P_S^2}{2\varepsilon_0} \beta \frac{w}{d}, \tag{2.5}$$

where

$$\beta = \frac{14\zeta(3)}{\pi^3} \frac{1}{1 + \chi \kappa_c}, \tag{2.6}$$

and $\zeta(n)$ is the Riemann zeta function. An analytic expression is obtained for the equilibrium domain width

$$w(d) = \sqrt{l_k d}, \tag{2.7}$$

where

$$l_k = \frac{2\varepsilon_0 \Sigma}{P_S^2 \beta} \tag{2.8}$$

is the Kittel length, which defines a characteristic length scale of the system. Equation (2.7) is known as Kittel's Law [1].

Beyond the Kittel regime, we can obtain the equilibrium domain width from the numerical solution to equation (2.1) for the full electrostatic expression in equation (2.4). In figure 3, we plot the domain width as a function of thickness both from the Kittel Law and equation (2.4) with numerical solutions, truncated at $n = 100$ terms. We use PbTiO$_3$ (PTO) and SrTiO$_3$ (STO) as examples of ferroelectric and paraelectric materials, respectively, in all of the plots in this paper, using suitable parameters.[1] PTO and STO are some of the most widely studied ferroelectric and paraelectric materials, respectively, both experimentally and theoretically, particularly in the context of FE/PE superlattices. The predictions of the model should be reliable in the Kittel regime, but other materials will better conform to the approximations of this model for single sub-Kittel thin films. The model is suitable, however, for strongly coupled PTO/STO superlattices, as for the situations described in [34,56].

---

[1]The following values of $d$ were used: $d_1 = 2$ nm, $d_2 = 1$ nm, $d_3 = 0.4$ nm, $d_4 = 0.105$ nm, $d_5 = 0.1$ nm, $d_6 = 0.99$ nm, $d_6 = 0.9$ nm. The values of the parameters used are: $P_S = 0.78$ C m$^{-2}$, $\Sigma = 0.13$ J m$^{-2}$, $\chi_\eta = 26$, $\kappa_a = 185$, $\kappa_c = 34$, $\kappa_s = 300$.

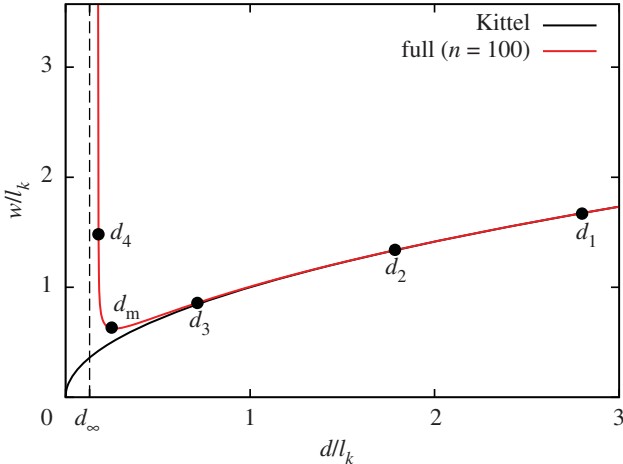

**Figure 3.** Equilibrium domain width as a function of thickness for an isolated thin film. The red curve shows the numerical solution using the full expression for the electrostatic energy, truncated at $n = 100$ terms. The solid black curve is the Kittel curve, scaled by the Kittel length: $w(d)/l_k = \sqrt{d/l_k}$. A number points are marked with black dots, which are referred to in figure 4 in order to show the evolution of domain width with thickness (scaled by the Kittel length). In particular, $d_m$ is the thickness at which the domain width is minimal and $d_\infty$ is the thickness at which the domain width diverges. The values of the parameters used are: $P_S = 0.78$ C m$^{-2}$, $\Sigma = 0.13$ J m$^{-2}$, $\kappa_a = 185$, $\kappa_c = 34$, $\kappa_s = 300$.

In figure 3, we see that the domain width follows Kittel's Law at large values of $d$, but, for decreasing $d$, $w$ reaches a minimum at $d_m$ and then diverges at $d_\infty$. We can understand this behaviour by studying the shape of the energy curves as a function of domain width and thickness, which is done in figure 4. The energy per unit volume associated with creating the domain walls, shown in red, is unaffected by the thickness of the film. The dashed grey lines show the electrostatic energy equation (2.4) at different thicknesses. We can see in each case that for small $w$, the energy is approximately linear in $w$, following Kittel's Law (equation (2.5)). As $w$ increases, Kittel's Law breaks down, and the curves begin to saturate to the monodomain electrostatic energy

$$\mathcal{F}_{\mathrm{mono}} = \frac{P_S^2}{2\varepsilon_0 \kappa_c}. \tag{2.9}$$

As $d$ decreases, the saturation of the electrostatic energy is realized earlier, and the minimum in total energy becomes shallower, eventually disappearing, the equilibrium domain width thereby diverging. We can visualize this by looking at the minima of the total energy curves as $d$ is decreased. The minima are marked with black dots on figure 4 and are also shown on the plot of $w(d)$ in figure 3.

The described deviation from Kittel's Law is sensitive to the system's parameters. In [7], an expression for $d_m$ was reported[2] of the form

$$d_m = 5\pi\Sigma\varepsilon_0 \frac{\kappa_c}{\chi} \frac{1}{P_S^2}, \tag{2.10}$$

where such dependence is explicit.

In figure 5, we show the effect of changing $\kappa_c$. Increasing $\kappa_c$ decreases the curvature of the electrostatic energy and also decreases the monodomain energy (the asymptotic energy for large $w$). By increasing $\kappa_c$ for a fixed value of $d$, the total energy minimum again becomes shallower and then disappears.

Although analytic solutions for the equilibrium domain width cannot be obtained using equation (2.4), we can obtain approximate solutions. Close to $d_m$, below which the width begins to diverge, we have

$$\left.\begin{array}{l} w(d) \cong \frac{\pi\chi}{2\sqrt{e}} d \exp\left(\frac{\pi^2}{8}\frac{\kappa_c}{\chi}\beta\frac{l_k}{d}\right) \\[2mm] d_m \cong \frac{\pi^2}{8}\frac{\kappa_c}{\chi}\beta l_k = \frac{\pi^2}{4}\Sigma\varepsilon_0\frac{\kappa_c}{\chi}\frac{1}{P^2}. \end{array}\right\} \tag{2.11}$$

Details of this approximation are given in appendix B and in [23]. In this approximation, $d_m$ has the same dependence on the system's parameters as equation (2.10), but the constant prefactor is different.

---

[2]The authors in [7] do not provide details on how this was obtained.

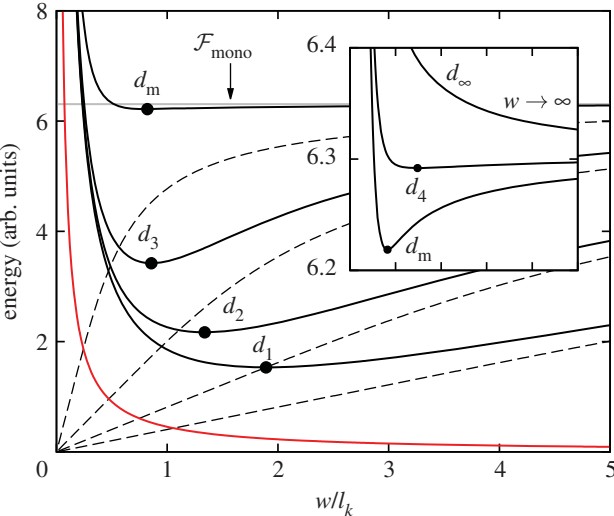

**Figure 4.** Energy as a function of domain width (scaled by the Kittel length) for the various values of $d$ introduced in figure 3. The red curve is the domain wall term. The black curves are the total energies for different values of $d$, and the dashed curves immediately beneath are the respective electrostatic energies at the same thicknesses (truncated at $n = 100$ terms). The minimum with respect to $w$ is indicated with a black dot. The inset shows the energy curves near where the equilibrium domain width diverges.

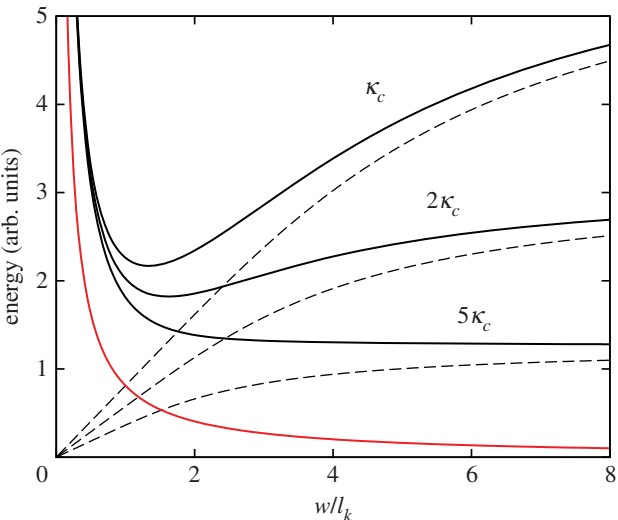

**Figure 5.** Energy domain width (scaled by the Kittel length) for a fixed value of $d$ and various multiples of $\kappa_c = 34$. The red curve is the energy cost of creating a domain structure. The black curves are the total energies for different values of $\kappa_c$, and the dashed grey curves immediately beneath are the respective electrostatic energies at the same thicknesses (truncated at $n = 100$ terms).

We can also obtain an analytic approximation to the domain width at all thicknesses by replacing equation (2.4) with a simpler expression which has the correct behaviour in the monodomain and Kittel limits

$$\mathcal{F}^*_{\text{elec}} = \frac{P_S^2}{2\varepsilon_0\kappa_c}\frac{1}{1 + (1/\kappa_c\beta)(d/w)}, \tag{2.12}$$

which clearly tends to equations (2.9) and (2.5) when $w/d$ is large and small, respectively. Using this, we get

$$\left.\begin{aligned} w(d) &= \frac{\sqrt{l_k d}}{1 - \kappa_c\beta\sqrt{l_k/d}} \\ d_m &= 4\kappa_c^2\beta^2 l_k \approx \frac{112\zeta(3)}{\pi^3}\Sigma\varepsilon_0\frac{\kappa_c}{\chi}\frac{1}{P_S^2}. \end{aligned}\right\} \tag{2.13}$$

Details of this approximation are given in appendix C. This approximation is of the same form as equation (2.10) but again with a different numerical prefactor. Equation (2.13) gives a good approximation to $d_m$, but overestimates the domain width near $d_m$. This is because, while equation (2.12) has the correct behaviour in the monodomain and polydomain limits, it underestimates the curvature in the intermediate region. In spite of this, the approximation predicts the correct dependence on the system's parameters.

Having understood the behaviour of the equilibrium domain width with thickness and the system's parameters, we proceed to investigate the effect of changing the surrounding environment of the thin film. For that purpose a more general expression for the electrostatic energies, similar to equation (2.4) is needed.

# 3. Generalized electrostatics

The electrostatic energies were obtained for the OL, SW and SL cases. The expressions, including their derivation, are shown in detail in appendix A. Some of the predictions of the model have been discussed previously in the literature [19,22–28]. To our knowledge, some of the SL results and all of the OL results are new. The results for all are presented and compared here.

## 3.1. Generalized Kittel Law

Taking the Kittel limit for the energies in equations (A 12) and (A 13), we obtain a generalization of Kittel's Law:

$$\left.\begin{aligned} w(d) &= \sqrt{l_k(\kappa_s)d} \\ l_k(\kappa_s) &= \frac{2\varepsilon_0 \Sigma}{P_S^2 \beta(\kappa_s)}, \end{aligned}\right\} \tag{3.1}$$

where $\kappa_s$ is the permittivity of the surrounding dielectric material. The generalization is introduced through the factor $\beta$

$$\left.\begin{aligned} \beta_{SW}(\kappa_s) &= \frac{14\zeta(3)}{\pi^3}\frac{1}{\kappa_s + \chi\kappa_c} \\ \beta_{SL}(\kappa_s,\,\alpha) &= \frac{1}{1+\alpha}\frac{14\zeta(3)}{\pi^3}\frac{1}{\kappa_s + \chi\kappa_c} \\ \beta_{OL}(\kappa_s) &= \frac{7\zeta(3)}{\pi^3}\left(\frac{1+\kappa_s+2\chi\kappa_c}{(1+\chi\kappa_c)(\kappa_s+\chi\kappa_c)}\right). \end{aligned}\right\} \tag{3.2}$$

The SL case has an additional dependence on $\alpha \equiv d_{PE}/d_{FE}$, the ratio of thicknesses of the paraelectric and ferroelectric layers. However, the energy cost of creating a domain wall is also renormalized by this prefactor, and thus, in the Kittel limit, the ratio $\alpha$ affects the energy scale but does not influence the behaviour of the domains. For each case in equation (3.2), equation (2.6) is recovered in the limit $\kappa_s \to 1$ (and $\alpha \to 0$ for the SL case).

The domain widths for the four different systems are plotted in figure 6. We can see that including the environment has the effect of shifting the curve upwards, but the square-root behaviour is unaffected. This makes sense physically: the paraelectric medium contributes to the screening of the depolarizing field. For higher dielectric constants, this contribution grows, meaning less screening is required by the domains, so there are fewer domains, and hence the width increases.

The SL and SW cases have the exact same behaviour in the Kittel limit. This is expected, since in the Kittel limit, the field in the superlattice loops in the paraelectric layers but does not penetrate through to neighbouring ferroelectric layers. In this regime, the coupling between the ferroelectric layers is weak, and the ferroelectric layers are essentially isolated from each other, tending to the SW case.

In [22], it was claimed that there should be a factor of two between the length scales of the OL and SW systems. From equation (3.2), we have

$$\frac{l_{k,OL}(\kappa_s)}{l_{k,SW}(\kappa_s)} = \frac{\beta_{SW}(\kappa_s)}{\beta_{OL}(\kappa_s)} = \frac{1+\chi\kappa_c}{1+\kappa_s+2\chi\kappa_c}. \tag{3.3}$$

When $\kappa_s \approx 1$, this is indeed true. However, when $\kappa_s$ is comparable to or larger than $\chi\kappa_c$, the approximation is not valid. For example, for PTO and STO, $\chi\kappa_c \sim 79$ and $\kappa_s = 300$ and can be as large

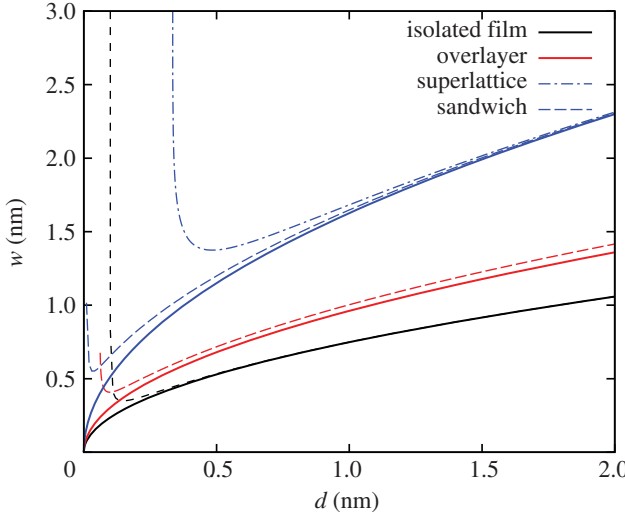

**Figure 6.** $w(d)$ for a thin film in a vacuum (black), the OL system (red), the SW (blue) and the SL system with $\alpha = 3$ (blue). The solid lines show the analytic solutions from the generalized Kittel's Law and the dashed lines show numerical solutions using the full expressions for the depolarizing energies. The SL and SW systems have identical square-root curves in the Kittel limit.

as $10^4$ at low temperatures, and the difference in the Kittel lengths becomes significantly larger than a factor of two.

## 3.2. Beyond Kittel: thin films

Although the square-root curve is simply shifted upwards after including the environment, the behaviour for thinner films is quite different. In figure 6, we can see that the thickness at which the domain width diverges is very sensitive to the dielectric environment. In figure 7, we plot the domain widths for various values of the dielectric permittivity of the substrate material, $\kappa_s$, for the OL and SW systems, each curve scaled by the relevant Kittel length, $l_k(\kappa_s)$. We see that $d_m$ decreases with increasing $\kappa_s$. In figure 8, we plot the critical thickness as a function of $\kappa_s$ to illustrate this effect. For the SW system, $d_m$ decreases more dramatically. This is expected, as there is screening on both sides of the thin film in the SW system.

We can understand the effect of the paraelectric permittivity on $d_m$ by examining the form of the electrostatic energy. For example, for the SW system

$$\mathcal{F}_{\text{elec}}^{\text{SW}} = \frac{1}{\kappa_s} \frac{8P_S^2}{\varepsilon_0 \pi^3} \frac{w}{d} \sum_{n \text{ odd}} \frac{1}{n^3} \frac{1}{1 + \chi(\kappa_c/\kappa_s)\coth\left((n\pi/2)\chi(d/w)\right)}. \tag{3.4}$$

This is equivalent to the electrostatic energy of the IF system, but with the overall energy and $\kappa_c$ both scaled by $\kappa_s$. As we know from equations (2.10) and (2.13) that $d_m \propto \kappa_c^{3/2}$, it is clear that $d_m$ should decrease with increasing $\kappa_s$.

## 3.3. Superlattice

For the SL system with $\alpha = d_{\text{PE}}/d_{\text{FE}} = 1$, we find that $d_m$ actually increases with the permittivity of the paraelectric layers, as shown in figure 9a, contrary to what happens for OL and SW. For small values of $\alpha$, the periodic boundary conditions of the superlattice make the electrostatic description very different from the OL and SW systems. When the paraelectric layers are thin, the depolarizing field penetrates through them and there is strong coupling between the ferroelectric layers. The superlattice acts as an effectively uniform ferroelectric material. The average polarization decreases with the permittivity of the paraelectric layers, and according to equation (2.10), $d_m$ increases.

For large spacings between the ferroelectric layers ($\alpha \gg 1$), the coupling between them becomes weak, the SW system being realized for $\alpha \to \infty$. This is illustrated in figure 9c, which is almost identical to figure 7b.

Interestingly, when $\alpha = \alpha_c \equiv \chi = \sqrt{\kappa_a/\kappa_c}$, $d_m/l_k(\kappa_s)$ is independent of $\kappa_s$. At this ratio, the dielectric permittivity of the spacer has no influence on the equilibrium domain structure, relative to the length scale given by $l_k(\kappa_s)$. This is shown in figure 9b. In figure 10, we plot $d_m$ as a function of $\kappa_s$ for

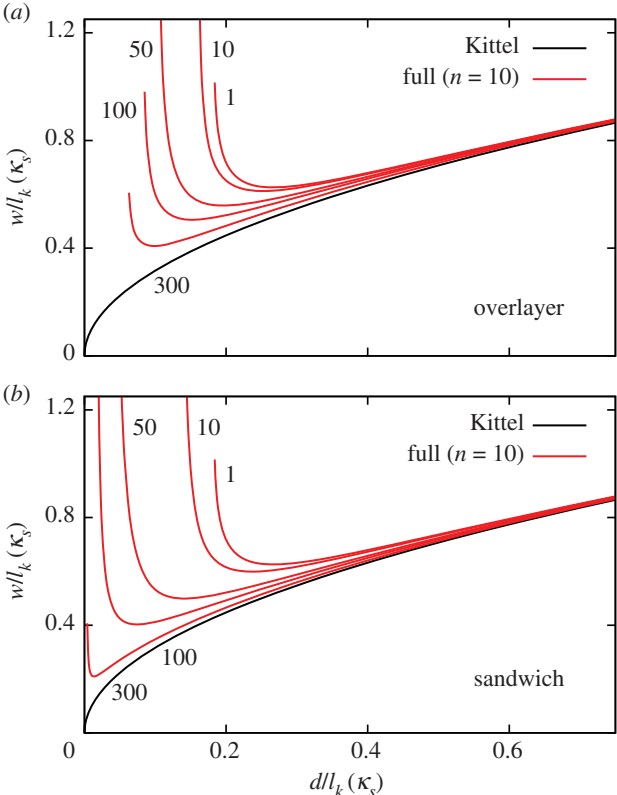

**Figure 7.** Domain widths as a function of thickness for various values of $\kappa_s$ for (a) the OL system and (b) the SW system. Each domain width and film thickness is normalized by the Kittel length for that value of $\kappa_s$.

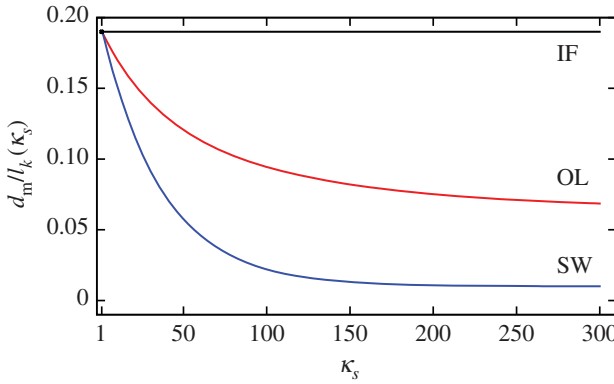

**Figure 8.** $d_m$ relative to the corresponding Kittel length as a function of dielectric permittivity of the substrate material for the OL (red) and the SW (blue) systems.

different values of $\alpha$. We see that when $\alpha > \alpha_c$, $d_m$ increases with $\kappa_s$, while it decreases for $\alpha < \alpha_c$, and remains constant when $\alpha = \alpha_c$. Thus, $\alpha_c$ represents a natural boundary between the strong and weak coupling regimes of superlattices.

The critical ratio $\alpha_c$ can be predicted from both the asymptotic and analytic approximations. Using the analytic approximation to the SL system (see appendix C), we have

$$\frac{d_m}{l_k(\kappa_s)} = 4(\kappa_c + \alpha^{-1}\kappa_s)^2 \beta(\kappa_s)^2$$

$$\propto \frac{\kappa_c}{\kappa_a} \frac{(1 + (\kappa_s/\alpha\kappa_c))^2}{(1 + (\kappa_s/\chi\kappa_c))^2} \frac{1}{P_S^2}. \tag{3.5}$$

From this, we can see that when $\alpha = \alpha_c$, the dependence on $\kappa_s$ vanishes.

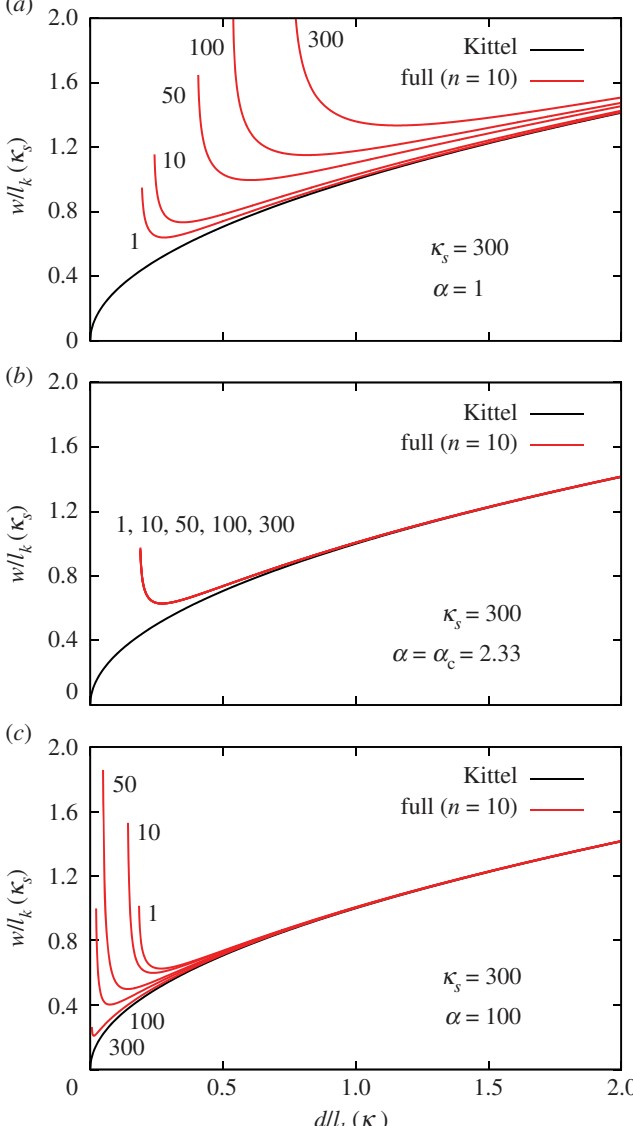

**Figure 9.** Domain width as a function of thickness for the SL system with (a) $\alpha = 1$, (b) $\alpha = \alpha_c$ (=2.33 for PTO/STO) and (c) $\alpha = 100$. Each domain width and film thickness is normalized by the Kittel length for that value of $\kappa_s$.

## 4. Discussion and conclusion

We have extended the continuum electrostatic description of an isolated ferroelectric thin film within Kittel's model to thin films surrounded by dielectric media and FE/PE superlattices. While some of the generalizations have previously appeared in the literature, a detailed comparison had not been done before. In doing so, we have understood how the surrounding dielectric materials influence the domain structure in the ferroelectric materials, both in the Kittel limit and beyond.

In the Kittel limit, the square-root behaviour is only affected in scale, the domain width increasing with dielectric permittivity, $\kappa_s$. This provides a useful correction to measurements of domain width with film thickness, as Kittel's Law for an IF typically underestimates domain widths. Beyond Kittel's regime, we found that increasing $\kappa_s$ decreases $d_m$, that is, the thickness for which the domain width is minimal.

For FE/PE superlattices, we found that $\kappa_s$ can either decrease or increase $d_m$, depending on the ratio of thicknesses, $\alpha = d_{PE}/d_{FE}$. We relate this to the different coupling regimes between the ferroelectric layers, as discussed in [19] for example. When $\alpha$ is large, the ferroelectric layers are weakly coupled, and the minimum thickness decreases with $\kappa_s$. When $\alpha$ is small, the ferroelectric layers are strongly coupled, and $d_m$ increases with $\kappa_s$. Remarkably, when $\alpha = \alpha_c \equiv \chi = \sqrt{\kappa_a/\kappa_c}$, the dielectric anisotropy of the ferroelectric layers, $d_m/l_k$ is unaffected by $\kappa_s$. In reality $d_m$ does change, since the Kittel length

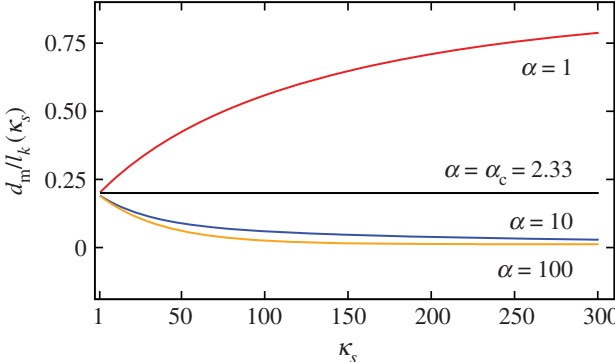

**Figure 10.** Critical thickness of the SL system as a function of $\kappa_s$ for several values of $\alpha$. Each value of $d$ is scaled by the appropriate Kittel length.

depends on $\kappa_s$, but the scaling is different above/below $\alpha_c$. The critical ratio $\alpha_c$ serves as a clear boundary between the strong and weak coupling regimes from an electrostatic viewpoint.

One important approximation in the Kittel-like model used here is the description of the polarization in the ferroelectric, assuming a dielectric linear-response modification of the spontaneous polarization $P_S$ (or using equation (2.3) instead of equation (2.2) as free energy term related to the polarization). Within this approximation, the system approaches a monodomain phase in a thin-limit regime in which the more complete treatment may predict $P = 0$. We investigate this possibility by considering a theory with equation (2.2) for the polarization, and equation (2.12) as the model electrostatic energy. We find that the polarization is zero for small thicknesses until

$$d_c = 27(\kappa_c \beta)^2 l_k \tag{4.1}$$

at which the polarization jumps to $P_S/\sqrt{3}$ and quickly saturates to $P_S$ [34]. Or, coming from $d > d_c$, the polarization decreases and the domain width increases, until at $d_c$, the ferroelectric material becomes paraelectric.

If $d_c < d_\infty$ the theory is unaffected, and the polydomain to monodomain transition would occur before the ferroelectric to paraelectric transition. Otherwise, the ferroelectric film becomes paraelectric without a polydomain to monodomain transition. For an isolated thin film of PTO, $d_m \sim 0.2 l_k$ and $d_c \sim 0.8 l_k$, meaning a ferroelectric to paraelectric transition takes place before the polydomain to monodomain transition. However, $d_c$ is also very sensitive to the environment of the film. For a sandwich system with a thin film of PTO between two regions of STO, again $d_c \gg d_m$. For strongly coupled FE/PE superlattices (small $\alpha$), however, $d_m$ increases with $\kappa_s$, and we would have $d_m \gg d_c$, and therefore the thin-limit behaviour presented above should be observable before the films becoming paraelectric.

The model described in this paper makes use of a number of significant approximations. Domains are typically not straight or of infinite length, and the domain structure may not be an equilibrium one ($A \neq 0, \pm 1$, see appendix A). In addition, the polarization gradients expected close to surface, interfaces and domain walls are better described within a Ginzburg–Landau theory, which will give significantly different predictions for ultrathin films, where complex structures such as polar vortices and skyrmions have been observed [57,58].

The comparative study offered in this work, however, gives the expected behaviour of ferroelectric/dielectric heterostructures within the simplest Kittel continuum model (continuum electrostatics for a given spontaneous polarization and dielectric response, plus ideal domain wall formation). While the domain width outside of Kittel regime may not be a realistic description for some materials, the values of $d_m$ predicted by this theory provide an estimate for when Kittel's Law breaks down. In particular, we have seen how the breakdown of Kittel's Law can be changed by the material parameters of the ferroelectric, as well as the surrounding environment. The described behaviours are already quite rich, and we think they represent a paradigmatic reference as basis for the understanding of more complex effects. In particular for superlattices, the strong to weak coupling regime separation based on this simplest model should be a useful guiding concept.

Data accessibility. The work in this paper has no data and the figures did not require any special code to generate. They were generated by minimizing the free energy equation (2.1) with electrostatic energies equations (2.4), (A 12) and (A 13) numerically in Mathematica. A Mathematica notebook containing the code used to generate all figures in this paper is included in the electronic supplementary material.

Authors' contributions. D.B. developed the theory and performed the computations. M.M.B. developed the theory. E.A. conceived of the presented idea. All authors discussed the results and contributed to the final manuscript.
Competing interests. We declare we have no competing interests.
Funding. D.B. acknowledges funding from the EPSRCCentre for Doctoral Training in Computational Methods for Materials Science under grant no. EP/L015552/1. The authors acknowledge funding from Spain's MINECO Plan Estatal Consortium grant no. FIS2015-64886-C5-1-P within the Excellence program (Excelencia): 'SIESTA for the theory of instabilities and transport in low-dimensional functional materials' consortium with five subgrants in five institutions. Principal Investigator (Coordinator): E.A.
Acknowledgements. The authors thank Pablo Aguado-Puente for helpful discussions.

# Appendix A. Electrostatics

Following [6], we obtained the expressions for the electrostatic energies of the OL, SW and SL systems. We present the derivation for the SL system, but the method also applies to the OL and SW systems, the only difference being the boundary conditions.

Consider a periodic array of ferroelectric and paraelectric layers as shown in figure 11. The spontaneous polarization of the ferroelectric layer has a 180° stripe domain structure with magnitude $\pm P_S$ and widths $W_+$, $W_-$. The unit cell of such a system is formed by one positive and one negative polarization domain in the $x$-direction, with period $W = W_+ + W_-$, and one ferroelectric and one paraelectric layer in the $z$-direction, with period $D = d_{FE} + d_{PE}$. As mentioned previously, we assume that the widths of the domain walls are infinitely thin. Thus, we write the spontaneous polarization as a Fourier series

$$P_S(x) = AP_S + \sum_{n=1}^{\infty} \frac{4P_S}{n\pi} \sin\left(\frac{n\pi}{2}(A+1)\right)\cos(nkx), \tag{A 1}$$

where $A = \frac{W_+ - W_-}{W}$ is the mismatch between the domains and $k = 2\pi/W$. We can see that the spontaneous polarization is split into a monodomain term, the average polarization $AP_S$, and polydomain terms in the infinite series. The polydomain limit is obtained when $A \to 0$, i.e. the domain widths are equal. The monodomain limit is obtained when $A \to \pm 1$, i.e. one of the domain widths tends to zero. To obtain the electric fields in the SL, we must first determine the electrostatic potentials. They satisfy the following Laplace equations:

$$\left.\begin{array}{l} \kappa_{ij}\partial_i\partial_j\phi_{II} = 0 \\ \kappa_s\nabla^2\phi_I = \kappa_s\nabla^2\phi_{III} = 0, \end{array}\right\} \tag{A 2}$$

where regions I, II and III are the different parts of the unit cell as shown in figure 11. Since the terms in (A 1) are linearly independent, we can treat the monodomain and polydomain cases separately. Clearly, the potentials must be even and periodic in $x$, so the general solutions to (A 2) are of the form

$$\left.\begin{array}{l} \phi_I(x,z) = c_0^1(z) + \sum_{n=1}^{\infty}\cos(nkx)(c_n^1 e^{nkz} + d_n^1 e^{-nkz}) \\[2mm] \phi_{II}(x,z) = c_0^2(z) + \sum_{n=1}^{\infty}\cos(nkx)(c_n^2 e^{nk\sqrt{\kappa_a/\kappa_c}z} + d_n^2 e^{-nk\sqrt{\kappa_a/\kappa_c}z}) \\[2mm] \phi_{III}(x,z) = c_0^3(z) + \sum_{n=1}^{\infty}\cos(nkx)(c_n^3 e^{nkz} + d_n^3 e^{-nkz}). \end{array}\right\} \tag{A 3}$$

In order to obtain the potentials, we must use the symmetries and boundary conditions of the system to determine the coefficients

$$\left.\begin{array}{l} \phi_I\left(\frac{d_{FE}}{2}\right) = \phi_{II}\left(\frac{d_{FE}}{2}\right) \\[2mm] \phi_{III}\left(-\frac{d_{FE}}{2}\right) = \phi_{II}\left(-\frac{d_{FE}}{2}\right) \\[2mm] \phi_I\left(\frac{D}{2}\right) = \phi_{III}\left(-\frac{D}{2}\right) \\[2mm] (\boldsymbol{D}_I - \boldsymbol{D}_{II})\cdot\hat{n} = 0 \\[1mm] (\boldsymbol{D}_{III} - \boldsymbol{D}_{II})\cdot\hat{n} = 0 \\[1mm] \phi_I(z) = -\phi_{III}(-z). \end{array}\right\} \tag{A 4}$$

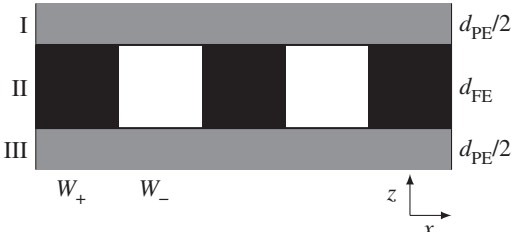

**Figure 11.** The geometry of a FE/PE superlattice. Regions I and III correspond to half of a paraelectric layer each and region II is the ferroelectric layer. The thicknesses of the layers are indicated on the right and $W_+$ and $W_-$ are the widths of the different domain orientations. The black squares are positive domains, with polarization $+P$ and the white squares are negative domains with polarization $-P$. The system is periodic in the horizontal and vertical directions, with periods $W = W_+ + W_-$ and $D = d_{FE} + d_{PE}$, respectively.

The first two conditions are obtained by matching the potentials at the interfaces. The third comes from imposing periodic boundary conditions on the unit cell. For the IF, OL and SW systems, the third condition would be replaced by

$$\left.\begin{array}{l} \lim_{z \to \infty} \partial_z \phi_I(x, z) = 0 \\[2mm] \lim_{z \to -\infty} \partial_z \phi_{III}(x, z) = 0. \end{array}\right\} \tag{A 5}$$

The fourth and fifth are obtained by matching the normal components of the displacement fields,

$$\left.\begin{array}{l} \mathbf{D}_I = \varepsilon_0 \kappa_s \mathbf{E}_I \\[1mm] \mathbf{D}_{II} = \varepsilon_0 \kappa \mathbf{E}_{II} + \mathbf{P_S} \\[1mm] \mathbf{D}_{III} = \varepsilon_0 \kappa_s \mathbf{E}_{III}, \end{array}\right\} \tag{A 6}$$

at the interfaces, and the final condition is obtained from the symmetry of the system under $z \to -z$.

After some algebra, we find that the potentials are given by

$$\left.\begin{array}{l} \phi_I(z) = -\dfrac{A P_S}{\varepsilon_0 \left[ \dfrac{\kappa_c}{d_{FE}} + \dfrac{\kappa_s}{d_{PE}} \right] d_{PE}} (z - D/2) \\[4mm] \quad - \sum\limits_{n=1}^{\infty} \alpha_n \beta_n \dfrac{\cos(nkx) \sinh(nk(z - D/2))}{\chi \kappa_c \cosh(nk\chi(d_{FE}/2)) + \kappa_s \coth(nk(d_{PE}/2)) \sinh(nk\chi(d_{FE}/2))} \\[5mm] \phi_{II}(z) = \dfrac{A P_S}{\varepsilon_0 \left[ \dfrac{\kappa_c}{d_{FE}} + \dfrac{\kappa_s}{d_{PE}} \right] d_{FE}} z \\[4mm] \quad + \sum\limits_{n=1}^{\infty} \alpha_n \dfrac{\cos(nkx) \sinh(nk\chi z)}{\chi \kappa_c \cosh\left(nk\chi \dfrac{d_{FE}}{2}\right) + \kappa_s \coth\left(nk \dfrac{d_{PE}}{2}\right) \sinh\left(nk\chi \dfrac{d_{FE}}{2}\right)} \\[5mm] \phi_{III}(z) = -\dfrac{A P_S}{\varepsilon_0 [\kappa_c/d_{FE} + (\kappa_s/d_{PE})] d_{PE}} (z + D/2) \\[4mm] \quad - \sum\limits_{n=1}^{\infty} \alpha_n \beta_n \dfrac{\cos(nkx) \sinh(nk(z + D/2))}{\chi \kappa_c \cosh(nk\chi(d_{FE}/2)) + \kappa_s \coth(nk(d_{PE}/2)) \sinh(nk\chi(d_{FE}/2))}, \end{array}\right\} \tag{A 7}$$

where

$$\left.\begin{array}{l} \alpha_n = \dfrac{4 P_S}{\varepsilon_0 n^2 \pi k} \sin\left(\dfrac{n\pi}{2}(A + 1)\right) \\[4mm] \beta_n = \dfrac{\sinh(nk\chi(d_{FE}/2))}{\sinh(nk(d_{PE}/2))}. \end{array}\right\} \tag{A 8}$$

The monodomain part of the potential has a zig-zag shape as expected, which is sensitive to the ratio of layer thicknesses and permittivities. The electrostatic energy of the system is obtained from

$$\mathcal{F}_{\text{elec}} = \frac{1}{2} \int \kappa_{ij} E_i E_j \, dx \, dz, \tag{A 9}$$

where the fields are the gradients of the potentials: $E = -\nabla \phi$. We integrate over the domain period in the $x$-direction and over both layers in the $z$-direction. Finally, the total electrostatic energy of the system is given by the second line of equation (A 12). The first line and equation (A 13) are the electrostatic energies of the SW and SL cases, respectively, obtained using the same method. In all cases, the energy is conveniently split into monodomain and polydomain parts. We can see that the monodomain parts for the OL and SW cases are identical to that of a thin film in a vacuum, as expected. We can also see that the polydomain part vanishes when $A \to \pm 1$, and the polydomain energy is obtained when $A \to 0$.

It will be useful for us to work in terms of energy *per unit volume*. For the OL and SW cases, we simply divide by the thickness of the thin film. For the superlattice, however, we must use the total volume of the unit cell. For convenience, we would like to work in terms volume of the ferroelectric layer. So we let

$$\left. \begin{array}{l} d = d_{\text{FE}} \\[4pt] \alpha = \dfrac{d_{\text{PE}}}{d_{\text{FE}}}, \end{array} \right\} \tag{A 10}$$

so that

$$\left. \begin{array}{l} d_{\text{PE}} = \alpha d \\[4pt] D = (1 + \alpha) d. \end{array} \right\} \tag{A 11}$$

The energies in equation (A 12) give a complete picture of the electrostatics of ferroelectric thin films and superlattices.

$$\left. \begin{array}{l} \mathcal{F}_{\text{elec}}^{\text{SW}} = \dfrac{P_S^2}{2\varepsilon_0 \kappa_c} \left( A^2 + \dfrac{16\kappa_c}{\pi^3} \dfrac{w}{d} \sum_{n=1}^{\infty} \dfrac{\sin^2\left(n\pi/2(A+1)\right)}{n^3} \dfrac{1}{\kappa_s + \chi\kappa_c \coth\left((n\pi/2)\chi(d/w)\right)} \right) \\[16pt] \mathcal{F}_{\text{elec}}^{\text{SL}} = \dfrac{1}{(1+\alpha)} \dfrac{P_S^2}{2\varepsilon_0 \kappa_c} \\[12pt] \quad \times \left( \dfrac{\kappa_c}{\kappa_c + \alpha^{-1}\kappa_s} A^2 + \dfrac{16\kappa_c}{\pi^3} \dfrac{w}{d} \sum_{n=1}^{\infty} \dfrac{\sin^2\left(n\pi/2(A+1)\right)}{n^3} \dfrac{1}{\chi\kappa_c \coth\left((n\pi/2)\chi(d/w)\right) + \kappa_s \coth\left((n\pi/2)\alpha(d/w)\right)} \right). \end{array} \right\} \tag{A 12}$$

For the substrate case, the energy is given by

$$\mathcal{F}_{\text{elec}}^{\text{OL}} = \dfrac{P_S^2}{2\varepsilon_0 \kappa_c} \left( A^2 + \dfrac{8\kappa_c}{\pi^3} \dfrac{w}{d} \sum_{n=1}^{\infty} \dfrac{\sin^2\left(n\pi/2(A+1)\right)}{n^3} \gamma_n^{-2} \Gamma_n \right), \tag{A 13}$$

where

$$\left. \begin{array}{l} \gamma_n = (\chi^2 \kappa_c^2 + \kappa_s) \sinh\left(n\pi\chi\dfrac{d}{w}\right) + \chi\kappa_c(1 + \kappa_s) \cosh\left(n\pi\chi\dfrac{d}{w}\right) \\[12pt] \Gamma_n = (\chi^2 \kappa_c^2 - \kappa_s)(1 + \kappa_s) - 4\chi^2 \kappa_c^2 (1 + \kappa_s) \cosh\left(n\pi\chi\dfrac{d}{w}\right) \\[12pt] \quad + (1 + \kappa_s)(3\chi^2 \kappa_c^2 + \kappa_s) \cosh\left(2n\pi\chi\dfrac{d}{w}\right) \\[12pt] \quad - 4\chi\kappa_c(\chi^2 \kappa_c^2 + \kappa_s) \sinh\left(n\pi\chi\dfrac{d}{w}\right) \\[12pt] \quad + \chi\kappa_c(1 + 2\chi^2 \kappa_c^2 + \kappa_s(4 + \kappa_s)) \sinh\left(2n\pi\chi\dfrac{d}{w}\right). \end{array} \right\} \tag{A 14}$$

It is important to check that the polydomain part of the energy reproduces the monodomain and Kittel energies in the appropriate limits. Letting $A = 0$, we have

$$\mathcal{F}_{\text{elec}}^{\text{SL}} = \dfrac{P_S^2}{2\varepsilon_0 \kappa_c} \left( \dfrac{16\kappa_c}{\pi^3} \dfrac{w}{d} \sum_{n \text{ odd}} \dfrac{1}{n^3} \dfrac{1}{\chi\kappa_c \coth\left((n\pi/2)\chi(d/w)\right) + \kappa_s \coth\left((n\pi/2)\alpha(d/w)\right)} \right), \tag{A 15}$$

ignoring the prefactor of $(1 + \alpha)^{-1}$. The monodomain limit is realized when $w \to \infty$. Using the expansion $\coth(ax) \sim 1/ax$ about $x = 0$, we get

$$
\begin{aligned}
\mathcal{F}_{SL} &\to \frac{P_S^2}{2\varepsilon_0(\kappa_c + \alpha^{-1}\kappa_s)} \frac{8}{\pi^2} \sum_{n \text{ odd}} \frac{1}{n^2} \\
&= \frac{P_S^2}{2\varepsilon_0(\kappa_c + \alpha^{-1}\kappa_s)},
\end{aligned}
\tag{A 16}
$$

since $\sum_{n \text{ odd}} 1/n^2 = \pi^2/8$. For the Kittel limit, $d/w \gg 1$. Using $\coth(x) \to 1$ for large $x$, we get

$$
\mathcal{F}_{elec}^{SL} \to \frac{P_S^2}{2\varepsilon_0} \frac{14\zeta(3)}{\pi^3} \frac{1}{\kappa_s + \chi\kappa_c} \frac{w}{d},
\tag{A 17}
$$

where we used $\sum_{n \text{ odd}} 1/n^3 = 7\zeta(3)/8$.

# Appendix B. Asymptotic approximation of the domain width in the ultrathin limit

Following the method in [23], we obtain an approximation to the equilibrium domain width behaviour in the ultrathin limit. For the IF system, total energy is approximately

$$
\mathcal{F} \cong \frac{\Sigma}{w} + \frac{8P_S^2}{\varepsilon_0\kappa_c\pi^2} \frac{1}{\xi} \sum_{n=0}^{\infty} \frac{1}{(2n+1)^3} \tanh\left(\frac{(2n+1)}{2}\xi\right),
\tag{B 1}
$$

when $\xi = \pi\chi\frac{d}{w} \ll 1$. Using

$$
\tanh\left(\frac{(2n+1)}{2}\xi\right) = \int_0^1 \partial_\lambda \left(\tanh\left(\frac{(2n+1)}{2}\xi\lambda\right)\right) d\lambda,
\tag{B 2}
$$

we get

$$
\begin{aligned}
\mathcal{F} &\cong \frac{\Sigma}{w} + \frac{4P_S^2}{\varepsilon_0\kappa_c\pi^2} \int_0^1 d\lambda \sum_{n=0}^{\infty} \frac{1}{(2n+1)^2} \frac{1}{\cosh^2(((2n+1)/2)\xi\lambda)} \\
&\approx \frac{\Sigma}{w} + \frac{16P_S^2}{\varepsilon_0\kappa_c\pi^2} \int_0^1 d\lambda \sum_{n=0}^{\infty} \frac{e^{-(2n+1)\xi\lambda}}{(2n+1)^2}.
\end{aligned}
\tag{B 3}
$$

From [23]

$$
\int_0^1 d\lambda \sum_{n=0}^{\infty} \frac{e^{-(2n+1)\xi\lambda}}{(2n+1)^2} = \frac{\pi^2}{8} - \frac{\xi}{4} \ln\left(\frac{e^p}{\xi}\right) + \mathcal{O}(\xi^3),
\tag{B 4}
$$

where $p = 1/2(3 + \ln(4))$. Thus, our approximation to the energy becomes

$$
\mathcal{F} \cong \frac{\Sigma}{w} + \frac{P_S^2}{2\varepsilon_0\kappa_c} + \frac{P_S^2}{2\varepsilon_0\kappa_c}\left(3 - \frac{8}{\pi}\chi\frac{d}{w}\ln\left(\Lambda\frac{w}{d}\right)\right),
\tag{B 5}
$$

where

$$
\Lambda = \frac{e^p}{\pi\chi}.
\tag{B 6}
$$

The first two terms are the domain energy and monodomain energy, and the third term is an asymptotic correction. Minimizing with respect to $w$, we get

$$
w(d) = \frac{\pi\chi}{2\sqrt{e}} d \exp\left(\frac{\pi^2}{8} \frac{\kappa_c}{\chi} \beta \frac{l_k}{d}\right).
\tag{B 7}
$$

The corresponding minimum width is

$$
d_m = \frac{\pi^2}{8} \frac{\kappa_c}{\chi} \beta l_k.
\tag{B 8}
$$

# Appendix C. Analytic approximation to the domain width

We can obtain an analytic approximation to the equilibrium domain behaviour if we replace the electrostatic energy with a simpler function which reproduces the monodomain and Kittel energies in the appropriate limits. For the IF system, we could use

$$\mathcal{F}^*_{\text{elec}} = \underbrace{\frac{P_S^2}{2\varepsilon_0 \kappa_c}}_{\mathcal{F}_{\text{mono}}} \frac{1}{1 + \frac{1}{\kappa_c \beta} \frac{d}{w}}. \tag{C 1}$$

When $w/d$ is very large, the second term in the denominator goes to zero and we get $\mathcal{F}^*_{\text{elec}} = \mathcal{F}_{\text{mono}}$. When $w/d$ is very small, the second term in the denominator dominates and we get $\mathcal{F}^*_{\text{elec}} = (P_S^2/2\varepsilon_0)\beta\frac{w}{d} = \mathcal{F}_{\text{Kittel}}$. This approximation can also be used for the OL and SW systems, since the extension to these systems is simply achieved via $\beta \to \beta(\kappa_s)$. For the superlattice, the energy in the monodomain limit is different

$$\mathcal{F}^{\text{SL}}_{\text{mono}} = \frac{1}{(1+\alpha)} \frac{P_S^2}{2\varepsilon_0(\kappa_c + \alpha^{-1}\kappa_s)}. \tag{C 2}$$

The prefactor $(1+\alpha)^{-1}$ scales the energy with the ratio of the layer thicknesses. The energy cost of creating a domain structure is also scaled by this prefactor. Now, the monodomain energy for a SL is similar to the case of a thin film, but with renormalized permittivity: $\kappa_c \to \kappa_c + \alpha^{-1}\kappa_s$. When $\alpha \to \infty$, the thin film expressions are recovered, so we can work with the SL system and the other systems can be recovered by taking $\alpha \to \infty$ and the correct choice of $\beta(\kappa_s)$.

The total energy for the SL system is

$$\mathcal{F} = \frac{\Sigma}{w} + \frac{\mathcal{F}^{\text{SL}}_{\text{mono}}}{1 + (x/w)}, \tag{C 3}$$

where

$$x = \frac{d}{(\kappa_c + \alpha^{-1}\kappa_s)\beta(\kappa_s)}. \tag{C 4}$$

Minimizing equation (C 3), we get

$$w(d) = \frac{\sqrt{l_k(\kappa_s)d}}{1 - (\kappa_c + \alpha^{-1}\kappa_s)\beta(\kappa_s)\sqrt{l_k(\kappa_s)/d}}. \tag{C 5}$$

Clearly, this expression has square-root behaviour for large $d$ (Kittel) and diverges for small $d$ (monodomain). The width diverges at

$$d_\infty = (\kappa_c + \alpha^{-1}\kappa_s)^2\beta(\kappa_s)^2 l_k(\kappa_s), \tag{C 6}$$

and has a minimum at

$$d_{\text{m}} = 4(\kappa_c + \alpha^{-1}\kappa_s)^2\beta(\kappa_s)^2 l_k(\kappa_s) = 4d_\infty$$
$$= 8\varepsilon_0(\kappa_c + \alpha^{-1}\kappa_s)^2\beta(\kappa_s)\Sigma\frac{1}{P_S^2}. \tag{C 7}$$

Interestingly, the relation $d_{\text{m}} = 4d_\infty$ is independent of system-specific parameters.

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
