## [Reviewer comments · Royal Society Open Science]

Review History

RSOS-201270.R0 (Original submission)

Review form: Reviewer 1

Is the manuscript scientifically sound in its present form?

Yes

Are the interpretations and conclusions justified by the results?

Yes

Is the language acceptable?

Yes

Do you have any ethical concerns with this paper?

No

Have you any concerns about statistical analyses in this paper?

No

Recommendation?

Accept with minor revision (please list in comments)

Comments to the Author(s)

This article studies a free energy model for the electrostatic problem of a finite-size ferroelectric layer under several boundary conditions (overlayer, sandwich, superlattice). The model describes the domain formation and domain widths of the ferroelectric layer as a function of the film thickness, and is a generalization of the Kittel-Mitsui-Furiuchi model. The authors describe in detail the model, compare it with previous models in the literature and assess its validity in different limits. The article is thorough, very interesting, and well written. It provides a better understanding of similar (but less general) models previously reported in the literature, making also a connection between systems with different boundary conditions, thus serving as a paradigmatic model. Also, the authors obtain some new results for overlayer and superlattice geometries. I certainly recommend it for publication, although I suggest some minor changes.

1) In the third paragraph of the introduction the authors state that in Ref.[22] it was claimed that a free-standing thin film on a substrate has the same electrostatic description as a thin film of half the width sandwiched between two paraelectric media. I believe this is not correct, it was claimed that it is equivalent to a thin film of half the thickness. I assume this is a typo, otherwise I think it deserves further explanation.

2) In the spirit of setting a paradigm for electrostatic models with this work, could the authors further clarify (possibly with an equation) the connection between the more complex form of $F_0(P)$ shown in Eq.(2) and its more simple form (mostly used throughout the text) in Eq.(3)?

3) When describing Fig. 5 the authors state that "By increasing κ_c for a fixed value of d , the total energy minimum again becomes shallower and then disappears". Could the authors make a brief comment on the physical implications of this finding?

4) When describing Eq. (15) the authors state that "For each case in Eq. (15), Eq. (6) is recovered in the limit $\kappa_s \rightarrow 1$." In the case of the superlattice this is true only for $\alpha=0$. I believe this should be stated explicitly.

5) κ_s is introduced in the main text in Eq. (14) (Section III. A) but is defined later (first paragraph of Section III. B). The authors should define it as soon as it is introduced.

6) The acronym "IF" is never defined, it should be defined the first time it is used (Section III. B). Does it correspond to "isolated film"?

7) In the second paragraph of the Discussion and Conclusion section the authors state that the domain width increases with dielectric permittivity in the Kittel limit. I believe this should be rephrased to emphasize that this discussion is restricted to systems with non-periodic boundary conditions, since in the superlattice case the above statement is only true for a particular range of values of α (as the authors have found, and as they discuss in the next paragraph).

8) In the first paragraph of Appendix A the authors present the derivation of the electrostatic energy for the SL system, the method applying also to the OL and SW systems. I do not believe that the derivation for the three systems should be included, since the three of them are very similar. I do believe, however, that the statement regarding the difference in the boundary conditions for the three geometries is a bit too vague ("the only difference being that the boundary conditions change from periodic to infinite."). I suggest that this sentence is rephrased for the text to be more precise and to allow the readers to reproduce the results for the cases not shown explicitly. Alternatively, the authors could write down explicitly the boundary conditions in a mathematical form.

9) In Appendix C the authors give the expressions for the electrostatic energies of the SW, SL and OL geometries in their generalized Kittel model. When describing them (page 9, line 57, first column) there are two minor mistakes that may misguide the reader: the authors state that "The first and third lines are the electrostatic energies of the SW and SL cases respectively", but there

are only two lines in Eq(30). Also, the SL case is the one for which the derivation has been presented step by step. I believe this minor mistake should be corrected for clarity (e.g. the sentence could read something like “The first line in Eq. (30) and Eq. (31) correspond to the electrostatic energies of the SW and OL cases respectively”, or, alternatively, Eq.(31) could become the third line of Eq. (30)).

10) In Appendix C, when discussing the effect of the prefactor $(1+\alpha)^{-1}$, the authors state that both the electrostatic energy and the “energy cost of creating a domain structure” are scaled by it. The authors then claim that “the equilibrium domain width will be unaffected by this prefactor, and we can neglect it”. I believe the very last statement (“we can neglect it”) is partially vague and somehow misleading, since in the next sentence the effect of the prefactor alpha on the (renormalized) permittivity is addressed, and later it is shown how it modifies other quantities like d_m and d_{inf} . I suggest either omitting it or rephrasing it with more precision.

11) In Fig. 4, could the authors show the saturation value for the energy (F_{mono}) for reference?

Finally, I attach a list of typos I found while reading the manuscript:

-Mixed use of American and British spelling of behavior/behaviour throughout the text.

-Page 1, line 46 - 48, first column: “however” is used twice in the last sentence of this paragraph for the same proposition.

-Page 4, line 40, second column: “systems parameters” → “system’s parameters”.

-Page 5, line 38, first column: “a more general expressions” → “a more general expression”

-Page 5, line 22, second column: “refactor” → “prefactor”

-Page 5, line 37, second column: “This expected” → “This is expected”

-Page 6, line 48, first column: “the width at which the domain width diverges” → “the thickness at which the domain width diverges”

-Page 6, line 41, first column: “of the substrate material, κ_s for the OL[...]” → “of the substrate material, κ_s , for the OL[...]”.

-Page 8, line 29, first column: “break down” → “breakdown”

-Ref [1]: “K. Charles” → “Kittel, C”

-Ref [55]: “M. Muoz-Basagoiti” → “M. Muñoz-Basagoiti”

-Page 9, line 45, second column: “we simply divide by the thickness the thin film” → “we simply divide by the thickness of the thin film”

-Page 11, line 17, first column: the polarization in the in-line formula should have an “S” subindex for consistency, “ P^2 ” → “ P_S^2 ”

Review form: Reviewer 2 (Mathew Dawber)

Is the manuscript scientifically sound in its present form?

Yes

Are the interpretations and conclusions justified by the results?

Yes

Is the language acceptable?

Yes

Do you have any ethical concerns with this paper?

No

Have you any concerns about statistical analyses in this paper?

No

Recommendation?

Accept as is

Comments to the Author(s)

The formation of ferroelectric domain structures in ferroelectric thin films and multilayers is a topic of active research. A key model that is frequently applied in these studies is the Kittel model, but it is most often applied without a great deal of thought about whether it should actually apply, where it might break down and how its applicability and predictions might differ depending on the environment surrounding the ferroelectric material.

In the present paper the authors have carried out a very rigorous investigation of these issues. The presentation of their results is excellent, with a good pedagogical style that explains well the background to the model. The authors have also made a good effort to present their results in a way that will be useful to experimentalists. The results that pertain to how the dielectric constant of the ferroelectric material shift the deviation from the Kittel law are particularly interesting.

Overall, considering the care the authors have taken in developing their model and presenting their results I am happy to recommend publication without revisions.

Decision letter (RSOS-201270.R0)

Dear Mr Bennett

On behalf of the Editors, we are pleased to inform you that your Manuscript RSOS-201270 "Electrostatics and domains in ferroelectric superlattices" has been accepted for publication in Royal Society Open Science subject to minor revision in accordance with the referees' reports. Please find the referees' comments along with any feedback from the Editors below my signature.

Please submit your revised manuscript and required files (see below) no later than 7 days from today's (ie 25-Sep-2020) date. Note: the ScholarOne system will 'lock' if submission of the revision is attempted 7 or more days after the deadline. If you do not think you will be able to meet this deadline please contact the editorial office immediately.

on behalf of Dr Robert Young (Associate Editor) and Miles Padgett (Subject Editor)
openscience@royalsociety.org

Reviewer comments to Author:

Reviewer: 1
Comments to the Author(s)

This article studies a free energy model for the electrostatic problem of a finite-size ferroelectric layer under several boundary conditions (overlayer, sandwich, superlattice). The model describes the domain formation and domain widths of the ferroelectric layer as a function of the film thickness, and is a generalization of the Kittel-Mitsui-Furiuchi model. The authors describe in detail the model, compare it with previous models in the literature and assess its validity in different limits. The article is thorough, very interesting, and well written. It provides a better understanding of similar (but less general) models previously reported in the literature, making also a connection between systems with different boundary conditions, thus serving as a paradigmatic model. Also, the authors obtain some new results for overlayer and superlattice geometries. I certainly recommend it for publication, although I suggest some minor changes.

1) In the third paragraph of the introduction the authors state that in Ref.[22] it was claimed that a free-standing thin film on a substrate has the same electrostatic description as a thin film of half the width sandwiched between two paraelectric media. I believe this is not correct, it was claimed that it is equivalent to a thin film of half the thickness. I assume this is a typo, otherwise I think it deserves further explanation.

2) In the spirit of setting a paradigm for electrostatic models with this work, could the authors further clarify (possibly with an equation) the connection between the more complex form of $F_0(P)$ shown in Eq.(2) and its more simple form (mostly used throughout the text) in Eq.(3)?

3) When describing Fig. 5 the authors state that "By increasing κ_c for a fixed value of d , the total energy minimum again becomes shallower and then disappears". Could the authors make a brief comment on the physical implications of this finding?

4) When describing Eq. (15) the authors state that "For each case in Eq. (15), Eq. (6) is recovered in the limit $\kappa_s \rightarrow 1$." In the case of the superlattice this is true only for $\alpha=0$. I believe this should be stated explicitly.

5) κ_s is introduced in the main text in Eq. (14) (Section III. A) but is defined later (first paragraph of Section III. B). The authors should define it as soon as it is introduced.

6) The acronym "IF" is never defined, it should be defined the first time it is used (Section III. B). Does it correspond to "isolated film"?

7) In the second paragraph of the Discussion and Conclusion section the authors state that the domain width increases with dielectric permittivity in the Kittel limit. I believe this should be rephrased to emphasize that this discussion is restricted to systems with non-periodic boundary

conditions, since in the superlattice case the above statement is only true for a particular range of values of α (as the authors have found, and as they discuss in the next paragraph).

8) In the first paragraph of Appendix A the authors present the derivation of the electrostatic energy for the SL system, the method applying also to the OL and SW systems. I do not believe that the derivation for the three systems should be included, since the three of them are very similar. I do believe, however, that the statement regarding the difference in the boundary conditions for the three geometries is a bit too vague ("the only difference being that the boundary conditions change from periodic to infinite."). I suggest that this sentence is rephrased for the text to be more precise and to allow the readers to reproduce the results for the cases not shown explicitly. Alternatively, the authors could write down explicitly the boundary conditions in a mathematical form.

9) In Appendix C the authors give the expressions for the electrostatic energies of the SW, SL and OL geometries in their generalized Kittel model. When describing them (page 9, line 57, first column) there are two minor mistakes that may misguide the reader: the authors state that "The first and third lines are the electrostatic energies of the SW and SL cases respectively", but there are only two lines in Eq(30). Also, the SL case is the one for which the derivation has been presented step by step. I believe this minor mistake should be corrected for clarity (e.g. the sentence could read something like "The first line in Eq. (30) and Eq. (31) correspond to the electrostatic energies of the SW and OL cases respectively", or, alternatively, Eq.(31) could become the third line of Eq. (30)).

10) In Appendix C, when discussing the effect of the prefactor $(1+\alpha)^{-1}$, the authors state that both the electrostatic energy and the "energy cost of creating a domain structure" are scaled by it. The authors then claim that "the equilibrium domain width will be unaffected by this prefactor, and we can neglect it". I believe the very last statement ("we can neglect it") is partially vague and somehow misleading, since in the next sentence the effect of the prefactor α on the (renormalized) permittivity is addressed, and later it is shown how it modifies other quantities like d_m and d_{inf} . I suggest either omitting it or rephrasing it with more precision.

11) In Fig. 4, could the authors show the saturation value for the energy (F_{mono}) for reference?

Finally, I attach a list of typos I found while reading the manuscript:

-Mixed use of American and British spelling of behavior/behaviour throughout the text.

-Page 1, line 46 - 48, first column: "however" is used twice in the last sentence of this paragraph for the same proposition.

-Page 4, line 40, second column: "systems parameters" → "system's parameters".

-Page 5, line 38, first column: "a more general expressions" → "a more general expression"

-Page 5, line 22, second column: "refactor" → "prefactor"

-Page 5, line 37, second column: "This expected" → "This is expected"

-Page 6, line 48, first column: "the width at which the domain width diverges" → "the thickness at which the domain width diverges"

-Page 6, line 41, first column: "of the substrate material, κ_s for the OL[...]" → "of the substrate material, κ_s , for the OL[...]"

-Page 8, line 29, first column: "break down" → "breakdown"

-Ref [1]: "K. Charles" → "Kittel, C"

-Ref [55]: "M. Muñoz-Basagoiti" → "M. Muñoz-Basagoiti"

-Page 9, line 45, second column: "we simply divide by the thickness the thin film" → "we simply divide by the thickness of the thin film"

-Page 11, line 17, first column: the polarization in the in-line formula should have an "S" subindex for consistency, " P^2 " → " P_S^2 "

Reviewer: 2
Comments to the Author(s)

The formation of ferroelectric domain structures in ferroelectric thin films and multilayers is a topic of active research. A key model that is frequently applied in these studies is the Kittel model, but it is most often applied without a great deal of thought about whether it should actually apply, where it might break down and how its applicability and predictions might differ depending on the environment surrounding the ferroelectric material.

In the present paper the authors have carried out a very rigorous investigation of these issues. The presentation of their results is excellent, with a good pedagogical style that explains well the background to the model. The authors have also made a good effort to present their results in a way that will be useful to experimentalists. The results that pertain to how the dielectric constant of the ferroelectric material shift the deviation from the Kittel law are particularly interesting.

Overall, considering the care the authors have taken in developing their model and presenting their results I am happy to recommend publication without revisions.

===PREPARING YOUR MANUSCRIPT===

- one version identifying all the changes that have been made (for instance, in coloured highlight, in bold text, or tracked changes);
- a 'clean' version of the new manuscript that incorporates the changes made, but does not highlight them.

This version will be used for typesetting.

===PREPARING YOUR REVISION IN SCHOLARONE===

Author's Response to Decision Letter for (RSOS-201270.R0)

See Appendix A.

Decision letter (RSOS-201270.R1)

Dear Mr Bennett,

It is a pleasure to accept your manuscript entitled "Electrostatics and domains in ferroelectric superlattices" in its current form for publication in Royal Society Open Science. The comments of the reviewer(s) who reviewed your manuscript are included at the foot of this letter.

===COVID-SPECIFIC TEXT -- WILL ONLY BE ADDED TO COVID-PAPERS BY THE EDITORIAL OFFICE===

COVID-19 rapid publication process:

We are taking steps to expedite the publication of research relevant to the pandemic. If you wish, you can opt to have your paper published as soon as it is ready, rather than waiting for it to be published the scheduled Wednesday.

This means your paper will not be included in the weekly media round-up which the Society sends to journalists ahead of publication. However, it will still appear in the COVID-19 Publishing Collection which journalists will be directed to each week (<https://royalsocietypublishing.org/topic/special-collections/novel-coronavirus-outbreak>).

If you wish to have your paper considered for immediate publication, or to discuss further, please notify openscience_proofs@royalsociety.org and press@royalsociety.org when you respond to this email.

===END OF COVID-SPECIFIC TEXT -- WILL BE REMOVED AS NECESSARY BY THE EDITORIAL OFFICE===

on behalf of Dr Robert Young (Associate Editor) and Miles Padgett (Subject Editor)
openscience@royalsociety.org

Appendix A

Statement on the Revision of Manuscript WM10136 Based on the Referee's Report

Daniel Bennett, Maitane Muñoz Basagoiti and Emilio Artacho

September 28, 2020

This statement concerns our revision of our manuscript RSOS-201270, entitled 'Electrostatics and domains in ferroelectric superlattices', based on the referees' reports.

Comments by Reviewer 1

This article studies a free energy model for the electrostatic problem of a finite-size ferroelectric layer under several boundary conditions (overlayer, sandwich, superlattice). The model describes the domain formation and domain widths of the ferroelectric layer as a function of the film thickness, and is a generalization of the Kittel-Mitsui-Furiuchi model. The authors describe in detail the model, compare it with previous models in the literature and assess its validity in different limits. The article is thorough, very interesting, and well written. It provides a better understanding of similar (but less general) models previously reported in the literature, making also a connection between systems with different boundary conditions, thus serving as a paradigmatic model. Also, the authors obtain some new results for overlayer and superlattice geometries. I certainly recommend it for publication, although I suggest some minor changes.

Thank you for taking the time to review our manuscript. We very much appreciate the care and attention to detail taken. We agree with all of the minor changes suggested and think they will help to make the manuscript clearer and overall more consistent. We will not respond to each one individually, but have implemented all of the suggested changes in the revised manuscript, highlighted in red.

Comments by Reviewer 2

The formation of ferroelectric domain structures in ferroelectric thin films and multilayers is a topic of active research. A key model that is frequently applied in these studies is the Kittel model, but it is most often applied without a great deal of thought about whether it should actually apply, where it might break down and how it's applicability and predictions might differ depending on the environment surrounding the ferroelectric material.

In the present paper the authors have carried out a very rigorous investigation of these issues. The presentation of their results is excellent, with a good pedagogical style that explains well the background to the model. The authors have also made a good effort to present their results in a way that will be useful to experimentalists. The results that pertain to how the dielectric constant of the ferroelectric material shift the deviation from the Kittel law are particularly interesting.

Overall, considering the care the authors have taken in developing their model and presenting their results I am happy to recommend publication without revisions.

Thank you for taking the time to review our manuscript.